# Influence of Sedentary Behaviour Interventions on Vascular Functions and Cognitive Functions in Hypertensive Adults—A Scoping Review on Potential Mechanisms and Recommendations

**DOI:** 10.3390/ijerph192215120

**Published:** 2022-11-16

**Authors:** Sneha Ravichandran, Suresh Sukumar, Baskaran Chandrasekaran, Rajagopal Kadavigere, Shivshankar K N, Hari Prakash Palaniswamy, Raghuraj Uppoor, Kayalvizhi Ravichandran, Meshari Almeshari, Yasser Alzamil, Ahmad Abanomy

**Affiliations:** 1Department of Medical Imaging Technology, MCHP, MAHE, Manipal 576104, India; 2Department of Exercise and Sports Science, MCHP, MAHE, Manipal 576104, India; 3Department of Radiodiagnosis and Imaging, KMC, MAHE, Manipal 576104, India; 4Department of Medicine, KMC, MAHE, Manipal 576104, India; 5Department of Speech and Hearing, MCHP, MAHE, Manipal 576104, India; 6Department of Radiodiagnosis and Imaging, K S Hegde Medical Academy, NITTE Deemed to Be University, Deralakatte, Mangaluru, Karnataka 575018, India; 7Department of Medical Imaging Technology, K S Hegde Medical Academy, NITTE Deemed to Be University, Derelakatte, Mangaluru, Karnataka 575018, India; 8Department of Diagnostic Radiology, College of Applied Medical Science, University of Hail, Ha’il 81442, Saudi Arabia; 9Department of Radiological Sciences, Department of Diagnostic Radiology, College of Applied Medical Science, King Saud University, P.O. Box 10219, Riyadh 11451, Saudi Arabia

**Keywords:** sedentary behaviour, prolonged sitting, hypertension, systolic blood pressure and diastolic blood pressure, mean arterial pressure, pulsed wave velocity, cerebral prefusion

## Abstract

Since the workplace has become desk-based and leisure time has become dominated by digital relaxation modes, the world is moving at a pace where physical activity has become a time-bound routine. The negative effects of extended sitting are a global concern since the workforce is becoming more desk based. There is a dearth of reviews that may link the knowledge on the effects of sedentary behaviour on hypertension and its accompanying damage to the brain and blood vessels and provide a future scope for the investigations connected to the relationship between sedentary behaviour and hypertension. Methods: Based on the database search and extensive research we did, we found studies that concentrated on the adverse effects of sedentary behaviour in association with blood pressure, cognitive decline and brain damage on adults. Results: We extracted 12 articles out of 20,625. We identified the potential adverse effects of sedentary behaviour, methods to reduce sedentary behaviour and the positive changes on health due to the interventions introduced. Sedentary lifestyle has shown a decline in human health. However, the visible symptoms presented later in life makes it very important to know the areas of decline and look for ways to curb the decline or procrastinate it.

## 1. Introduction

Owing to a great increase in computer-based jobs in modern workplaces, a substantial increase in sedentary behaviour has become evident across modern workspaces [1] Contemporary evidence claims a strong association between the occupational sedentary behaviour and the early cardiometabolic disease risk [2]. Empirical evidence claims unfavorable vascular effects associated with prolonged sedentary times [3].

Roughly around 2 billion people in the world are diagnosed with hypertension. It is one of the most common types of metabolic diseases causing various health hazards [4]. Higher blood pressure affects various tissues and organs like large and small arteries, eyes, heart, liver, kidneys and many more. Researchers around the world are working on care and treatment for vascular dysfunction, arterial stiffness, dyslipidemia, hypertensive retinopathy, choroidopathy, optic neuropathy and other systematic diseases associated with hypertension [5]. The main concern is that these diseases only come to light at later stages and research is required to identify them at earlier stages. Vascular changes like a drop in blood velocity and an increase in blood vessel thickness are a few basic changes that can shed light on future instances of renal damage, cognitive decline, deduction in brain volume and atrophy. When speaking about effects of hypertension on the brain, the most common changes that we see is decreased cerebral blood flow, decreased cerebral perfusion, increased white matter intensities (hyperintensities) and decreased gray matter [6,7].

Hypertension is directly related to cerebral autoregulation which is a process that maintains the cerebral blood flow during variations in blood pressure. The autoregulation in normotensive individuals is between 50–150 mmHg mean arterial pressure (MAP), whereas this range gets shifted in hypertensive individuals. Impaired autoregulation can cause deep white matter ischemic diseases, cognitive impairment, and vascular impairment in late adulthood [8].

Sedentary behaviour is defined as “any waking behaviour in a reclining, seated or lying position which requires low energy expenditure which is less or equal to 1.5 MET”. This makes it stand apart from physical inactivity [9]. In a desk-based job setup, employees are expected to work in a closed framework, where increased deadlines and work pressure unknowingly push them to remain seated for long hours. This could replace the time spent more actively and could be the precursor for the early cardiometabolic disease risks.

In the case of employees with hypertension, the chances of increased endothelial dysfunctions are higher than those of non-hypertensive employees due to sedentary behavior [10]. Early lab-based trials claim a positive association of improved endothelial functions in hypertensive individuals when the prolonged sitting was interrupted periodically [11,12,13].

The potential mechanism that are underpinning the endothelial dysfunction, due to sedentary behaviour and physical inactivity, are increased systolic and diastolic blood pressure, altered shear rate, lower flow-mediated dilation (FMD) and altered vascular shape [14,15]. Anecdotal evidence claims a better chance of reversing the unfavorable vascular changes with the resumption of physical activity [13].

The potential individual and environmental ways of interrupting sedentary behaviour and administering movement breaks, which may produce favorable vascular and cognitive outcome, are being explored. Mounting evidence has investigated the influence of the “Sit Stand workstation”, Sit—Stand transition, prompts, nudges, text messages and community sports [2,16,17]. However, the influence of the above interventions on endothelial functions in hypertensives remains unexplored. Furthermore, the evidence regarding the efficacy of the various strategies to interrupt prolonged sitting on the endothelial dysfunction in hypertensive office workers remains unconsolidated.

## 2. Materials and Methods

This review is based on the Arksey and O’Malley framework [18]. The following steps were followed: (1) identifying the research question, (2) identifying relevant studies to answer those research questions, (3) selection of study, (4) charting of data and (5) reporting and summarizing of result [19].

Identifying the research question.

We articulated this scoping review into three main research questions based on topic, population and outcomes. The target population included adults with hypertension and sedentary behaviour. Table 1 illustrates the research question and the outcome of interest.

### 2.1. Identification of Relevant Studies

#### 2.1.1. Search Strategies

The systematic search was conducted in five peer review journal databases (Scopus, PubMed, Web of Science, Ovid, and Science Direct) and studies based on randomised control trials conducted worldwide on hypertension, Sedentary Behaviour (SB) and Physical Activity (PA). The search timeline was between 1964 and 2022. Table 2 refers to the MeSH terms used with the necessary Boolean operators: “AND”, “OR” and “NOT”—with the appropriate wildcards. The search was started in March 2022 and concluded on 17 May 2022. Once the final set of searches was conducted in the presence of a senior librarian, an assessment of inclusion was conducted after 20 May 2022. The inclusion criteria were articles that provided information on the effects on Sedentary Behaviour and hypertension, and concentrated on related changes in brain and blood vessels. We also focused on the improved body changes due to physical activity interventions. Articles which were not in English, not full text, not open access, conference papers or under review were excluded. The study flows illustrated in Figure 1.

#### 2.1.2. Charting of Data

Based on the articles collected, we tabulated the components as authors, year, the country the research was conducted in, objectives, participants and activity break. Activity break is further divided into type of activity, mode, intensity and duration of the break. Furthermore, the details on control group, washout period and key findings are included related to hypertension, SB and PA. Table 3 showcases the tabular data on SB, PA on hypertension and its based intervention.

## 3. Results

Based on the search results found from 5 indexed databases, out of 22,371 articles, only 12 articles were found to provide answers to research questions based on SB, hypertension, its adverse effects and potential interventions to break SB, as shown in Figure 1. Most studies were conducted in the USA (N = 4, 33.3%), followed by Australia (N = 3, 25%). Different European countries had a share of studies, such as Spain (N = 1), Portugal (N = 1), UK (N = 1), Netherlands (N = 1, 33.3%) and Canada (N = 1, 0.84%). We did not encounter any studies conducted in developing or underdeveloped countries such as Asian, African and South American countries. Three studies specifically focused on vascular effects, four on cognitive and perfusion effects, and five on blood pressure and hypertension alteration. All these studies focused on SB and methods of breaking it.

### 3.1. SB and Vascular Effects

We found that when PA decreased, changes in vascular FMD, NMD, arterial stiffness, plasma nitric oxide level, intima-to-media thickness, and PWV were significant [22]. In this study, we observed that sitting for longer hours increased the carotid ankle PWV by 0.27 m/s and carotid-femoral as well as carotid radial PWV by 0.03 m/s. Similar results were also observed by [25] where the PWV was 0.03 m/s in SB.

### 3.2. SB and Cognitive Changes and Perfusion Effects

Evidence shows that SB significantly affects the brain. Based on these studies, it was found that there was a decrease in brain blood perfusion, alteration in z-scores and BDNF levels. On the other hand, there was evidence of decreased intima to media thickness in cerebral artery, cerebral auto regulation and middle cerebral artery velocity. In this study by [28], they observed that sitting for 4 h straight has decreased the diameter of common carotid artery by 5% and cerebral autoregulation from 39.16 phase in degrees to 35.83 phased in degrees when measured using very low frequency.

### 3.3. SB and Blood Pressure

Studies have shown that SB increases the DBP and SBP. They also showed significant changes in hypertensive subjected specifically. The authors in [30] observed that there was an increase in daytime ambulatory blood pressure with 2.8 mmHg in systolic and 1.9 mm when there was a prolonged sitting of 4 h in hypertensive adults.

### 3.4. SB Interventions and Blood Pressure

Based on these twelve articles, five studies implemented standing, four depicted moderate-to-high-velocity exercise, two suggested walking, and one suggested calf raises [24]. When observing the duration allotted for the intervention to be performed, most suggested a break of either 2 to 3 min every 30 min of SB. Few suggested early morning exercise for 1 h and additional breaks during work hours, whereas a few suggested a 4 min break for every 120 min of SB. Few studies have been conducted with longer time intervals, ranging from six weeks to three months.

For the standing activity, there were sit and stand workstations set up in the office or motivational messages promoting standing and moving around were used. One of the studies used a lightweight box folding activity for the upper limb in the standing posture to break the SB.

Different types of exercise, such as aerobic routine, ACMS guidelines and morning exercise routine, have been introduced. Two articles suggested walking during the working period, such as treadmill walking or light-intensity walking. One study was based on the use of calf-raise exercises in a sitting posture at a certain speed and observed the changes.

### 3.5. Effects of PA on BP

When observing the effects of PA on vascular changes, we found evidence of increased vertebral blood flow volume, improved ankle PWV, a better FMD/NMD ratio and decreased intima-to-media thickness. There was a decrease in carotid artery intima to media thickness by 6.4%, plasma nitric oxide level increase by 76.6% and FMD to NMD ratio increased by 36.2% [25]. 

Based on cognitive functional changes, PA improved brain perfusion, decreased cerebral artery thickness, increased serum BDNF and z-scores, increased MCA velocity and improved cognitive memory. There was an increase in the cerebral autoregulation phase in hypertensive adults who were subjected to intermittent walking from 41.93 to 46.91 [28]. 

Inverse effects were also observed based on blood pressure. An increase in PA decreased DBP and SBP in hypertensive subjects, except in one article. In one study, the effects of stopping PA after three months of active PA routine showed evidence of degradation of vascular and cognitive changes, but the rate of degradation was less than that in the pre-intervention stages [23].

## 4. Discussion

Based on the results, our review shed light on the aftereffects of sedentary behaviour in association with the hypertension individuals had to concur. We observed a slow decline in vascular function which indirectly affected cognitive function. With this, we have listed the negative effects of SB on hypertensive individuals

### 4.1. Vascular Changes

Ref. [21] concluded that in an SB scenario, with an increase in blood pressure, there was a significant decrease in vertebral blood flow volume. In 2014, Feairheller showed that intima-to-media thickness decreased by 6.4% and the FMD-to-NMD ratio increased by 36.2% in hypertensive individuals [25]. In a study conducted by Barone Gibbs et al. in 2017, it was observed that the carotid-ankle PWV decreased in subjects sitting for 3.40 h straight [22]. 

### 4.2. Cognitive Changes

None of the above tabulated articles reviewed directly describe the cognitive functional changes. However, the pre-aspects of cognitive decline like blood perfusion to the brain, effects on the working memory and presence of BDNF in serum were discussed.

Stoner et al., in 2019, conducted a study conducted on 20 healthy men and women and showed that being sedentary for 3 h straight decreased cerebral blood perfusion [24]. SB also decreased working memory and serum BDNF levels, as reported by [26]. As a precursor of these studies [28], conducted a study to observe the acute cerebra blood flow and cerebral auto regulation in the sedentary individuals by setting up a working simulation and made the group sit for four hours straight. The team observed that sitting for longer hours decreased cerebral autoregulation and middle cerebral artery blood velocity.

### 4.3. Alteration in Blood Pressure

With SB, there is an alteration in the SBP and DBP, as shown by Barone Gibbs B et al. They recruited 25 obese subjects with pre-stage 1 hypertension and made one group sit for 3.40 h, uninterrupted, to find an increase in DBP and mean arterial pressure [22]. This study is supported by an article dating back to 2000, where Cooper et al. studied 90 desk-bound workers to prove that there was an increase in blood pressure when they were sedentary [30].

With all these effects of SB on hypertensive individuals, studies have been conducted worldwide to reduce SB and increase PA. Over the years, many interventions have been introduced and studies on their effects have been conducted. Interventions, such as walking, standing, aerobics, morning exercise, calf raising, and running, were performed.

### 4.4. Standing

Most of the studies used standing as the better mode of PA. Barone Gibbs B et al. suggested using sitting and standing alternative routines for over 5 to 14 days with a switch of position for every 30 min. The team observed that there was a significant decrease in DBP and MAP [22]. To study this effect on pre and stage 1 hypertensive individuals who were non-medicated with anti-hypertensive drugs, Perdomo SJ et al., in 2019, made a sitting and standing routine for 30 min to switch position and observed that blood pressure was significantly decreased after the intervention [21]. Two studies conducted in the year 2018 with standing as the main intervention. Stephens SK et al. from Australia and Antle DM et al. from Canada studied desk-based workers and observed that having to stand for around 30 min has increased the mean blood flow by 77%, made the participants more active and observed that there was a decrease in the lower limb discomfort as well [27,29].

### 4.5. Walking

The next most common intervention was walking, which may be treadmill walking for 2 to 3 min for every consecutive hour or 30 min of sitting [13,28,31] Stephens et al., in 2018, studied outdoor walking for 30 min. All studies concluded that walking lowered blood pressure, increased working memory and serum BDNF, and increased middle cerebral artery velocity and cerebral autoregulation [27] The study by Carter et al. also observed that the increase in arterial velocity was significant when the participants walked for two minutes [28].

### 4.6. Calf Raise

To maintain work psychology and reduce distraction, yet break the SB, Stoner et al. (2019) used the calf raise routine as an intervention. They were instructed to sit uninterrupted, except for a calf-raise routine at 12 raises per minute for every 20 min of sitting. This study observed that the calf raise routine is less effective and can only help to preserve vascular functions and showed no significant improvement in cerebral perfusion [24].

### 4.7. Moderate to Vigorous Exercise

The last group of interventions is a moderate-to-vigorous exercise routine, such as aerobic or ACSM guideline-based routine. All these studies reported similar results, such as an increase in PA, making participants more active [20,25,30]. A study by Feairheller et al. showed that the carotid artery intima-to-media thickness was decreased by 6.4% and plasma nitric oxide level was increased by 76.6%, but there was no improvement in blood pressure.

One of the most interesting findings that we observed was in a Portuguese study conducted in 2018 by Leitao et al., where the ACSM guideline-based exercise routine was introduced for 3 months in hypertensive working women aged between 60 and 70 years old. After the intervention, the routine was stopped for the next three months, and data were collected to observe the recoil of vascular changes. During the exercise period, blood pressure was improved, and vascular functions were well maintained. However, during the washout period, the recoil of vascular functional decline was observed, but the rate of decline was much slower than that before the initiation of PA [23].

## 5. Conclusions

With this, we reviewed articles and collected evidence that observed the ill effects of SB on vascular and cognitive health in hypertensive populations and looked at ways to reconsider PA as the best method to improve vascular and cognitive health. In addition, we looked at how different types of PA can produce positive results for better health. We had a few limitations that are worth noting: (1) Since the main aim of this review was to provide scope for further systematic review and research to determine the long-term effects of SB and PA on hypertensive individuals, we did not study the risk of bias in these studies. (2) We only concentrated in this review on the working population, where the effects of SB on hospitalised, paralysed, and bedridden individuals were excluded which could be a scope for further research and review.

To conclude, SB may seem to be a relaxing and ineffective lifestyle, but deep inside, it causes a long turn effect which can only result in late life cognitive and metabolic decline. Adding more PA can reverse these effects and improve the current health status. The results summarised in this review can help promote a PA-induced work setup in which employees’ physical and cognitive health are considered.

## Figures and Tables

**Figure 1 ijerph-19-15120-f001:**
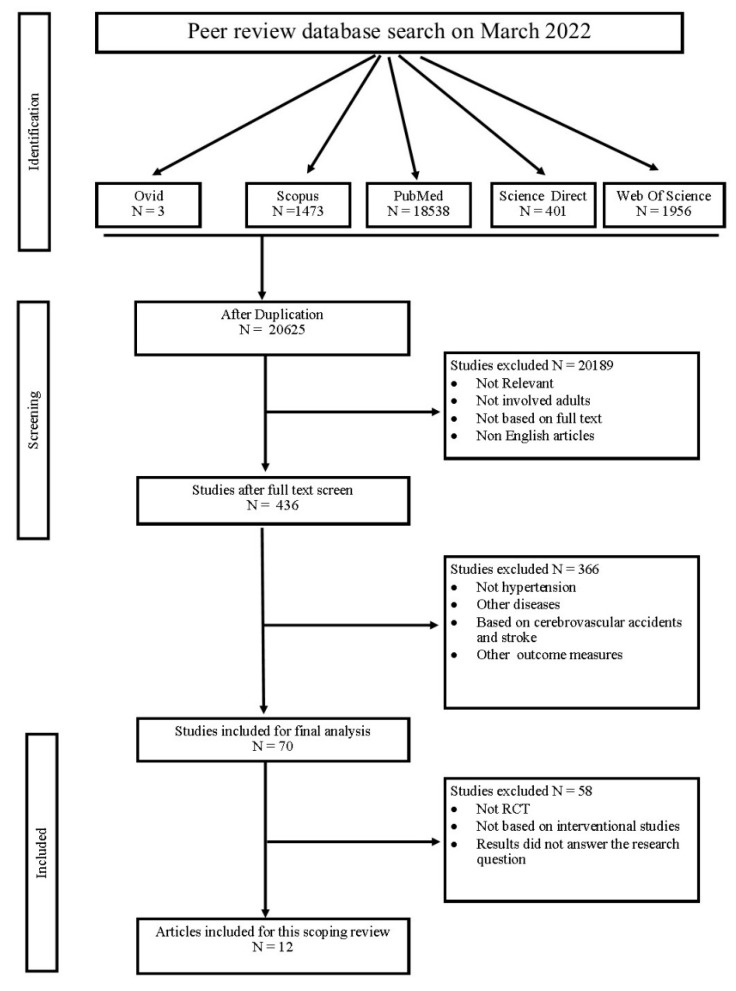
Data extraction flow chart.

**Table 1 ijerph-19-15120-t001:** Research questions and designated outcomes.

Research Question	Outcome of Interest
What are the vascular changes in hypertensive people observed during SB	Ventricular ejection, arterial stiffness Arterial pulse wave, FMD, carotid artery intima-media thickness
What are the cognitive changes observed during SB	Decreased cerebral blood flow, cerebral perfusion, decrease in gray and white matter, lower brain volumes and atrophy
What are the different interventions conducted to break the SB	Sit-stand workstation, standing, combination of sitting—standing—exercise, aerobics, JITAI—motivational PA based mobile messages, mobile based fitness apps, walking/treadmill walking, gentle jogger, calf raise.

Abbreviations: SB—Sedentary Behaviour, PA—Physical Activity, FMD—Flow Mediated Dilation.

**Table 2 ijerph-19-15120-t002:** MeSH terms.

Patient	Intervention	Comparison	Outcomes
**Desk based workers**	Movement breaks	Sedentary behaviour	Regional vascular functions
**Adults**	Interrupting prolonged sitting	Desk bound	FMD
**Software developers**	Physical activity	Hypertension SBP & DBP	Shear rate (SR)
**Office workers**	Micro-breaks	Arterial hypertension	Baseline diameter
**Receptionist**		Physical inactivity	Pulse wave velocity (PWV) arterial stiffness
**Typists**			Blood flow velocity

Abbreviations: SBP—Systolic Blood Pressure, DBP—Diastolic Blood Pressure, FMD—Flow Mediated Dilation.

**Table 3 ijerph-19-15120-t003:** Study classification and its characteristics.

Author, Year	Objective	Participants	Intervention				Control Group	Washout if Any	Key Findings
			Intervention	Intensity	Mode	Frequency			
Aguirre, 2020, Spain	“To analyze the interaction between aerobic exercise training, physical activity, sedentary behaviour and sleep quality in obese adults with primary hypertension” [20]	N = 218 Obese participants with hypertension	Exercise	Moderate to high velocity	Aerobic exercise EXERDIET—HTA	2 days per week for 16 weeks	30 healthy participants with no history of hypertension	Nill	Increased in PA from 117 min to 188 min and significant decrease in SB was observed
Perdomo, S.J.; Gibbs, B.B.; Kowalsky, R.J.; Taormina, J.M.; Balzer, J.R., 2019, USA	“To evaluate the effects of alternating standing and sittng versus prolonged sitting on cerebral blood flow velocity” [21]	N = 25 Participants with pre to stage 1 hypertension who are on non medication treatment	Sitting and standing	Low	Standing	30 min	Participants who are continuously sitting	Nill	Increase in Hypertension and BMI—decreases vertebral blood flow volume After the intervention—improvement in hypertension and decreased BMI
Barone Gibbs, B.; Kowalsky, R.J.; Perdomo, S.J.; Taormina, J.M.; Balzer, J.R.; Jakicic, J.M., 2017, USA	“To study whether intermittent standing improves vascular health is unclear. We aimed to test whether using a sit–stand desk could reduce blood pressure (BP) and pulse wave velocity (PWV) during a simulated workday” [22]	25 Obese subjects with pre-to-Stage 1 hypertension with SBP of 120–159 mmHg or DBP of 80–99 mmHg, and BMI of 25.0 to less than 40.0 kg/m2	Sitting for3.40 h	low	Standing	30 min	Group sitting continuously for 3:40 h	5–14 days	decrease DBP, MAP, and carotid–ankle PWV in standing-sitting subjects
Leitão, L.; Marocolo, M.; Souza, H.L.R.; Arriel, R.A.; Vieira, J.G.; Mazini, M.; Louro, H.; Pereira, A., 2021, Portugal	“To investigate the effects of a multicomponent training and detraining period in older women with hypertension” [23]	45 Working women aged between 60–70 years of age and hypertensive	ACSM guidelines for exercise prescription	high	Exercise routine	1 h for 3 months	Routine lifestyle baseline data before the interventions	3 months post intervention	Nine months of multicomponent exercise were sufficient to improve functional capacity and promote benefits in blood pressure, although was not sufficient to allow BP to reach the normal values of older women. The three month DT period without exercise caused the reversal of BP improvements but maintained the functional capacity of older women.
Stoner, L.; Willey, Q.; Evans, W.S.; Burnet, K.; Credeur, D.P.; Fryer, S.; Hanson, E.D., 2019, USA	“In young, healthy adults, (a) does exposure to acute prolonged (3 h) sitting lead to decreased cerebral perfusion and executive function? and (b) does breaking up prolonged sitting, using intermittent calf raise exercises, prevent changes in cerebral perfusion and executive function” [24]	20 Healthy young men and women	Sitting for 3 h	low	Calf raise	2 min of calf rise—10 calf rise at the speed of 12/min for every 20 min of sitting	Uninterrupted sitting for 3 h	Not applicable	Sitting decrease both cerebral perfusion and executive function. Simple strategies, such as fidgeting or calf raises, which have been reported to preserve vascular function in the legs, appear not to be sufficient to benefit cerebral perfusion or executive function.
Feairheller, D.L.; Diaz, K.M.; Kashem, M.A.; Thakkar, S.R.; Veerabhadrappa, P.; Sturgeon, K.M.; Ling, C.Y.; Williamson, S.T.; Kretzschmar, J.; Lee, H.; Grimm, H.; Babbitt, D.M.; Vin, C.; Fan, X.X.; Crabbe, D.L.; Brown, M.D., 2014, USA	“To study the improvements in NMD, FMD, plasma nitric oxide, carotid artery intima to media thickness, blood pressure, arterial blood pressure in sedentary African Americans” [25]	N = 26 Participants were African Americans adults	Exercise	Moderate to high velocity	Aerobic exercise	20 min of exercise for 3 times a week	Baseline data collected pre intervention	36–48 h	Decrease in carotid artery intima to media thickness by 6.4% Plasma nitric oxide level increase by 76.6% FMD to NMD ratio increased by 36.2% No improvement in blood pressure
Wheeler, M.J.; Dunstan, D.W.; Ellis, K.A.; Cerin, E.; Phillips, S.; Lambert, G.; Naylor, L.H.; Dempsey, P.C.; Kingwell, B.A.; Green, D.J., 2019, Australia	“To study if acute bout of exercise would reduce BP during an 8-h period, relative to prolonged uninterrupted sitting and that the BP reduction after acute exercise would be further enhanced by subsequent exposure to intermittent breaks in sitting” [13]	67 Men (n = 32) and postmenopausal women (n = 35) with body mass index, ≥25 to <45 kg/m^2^	Sitting for 8 h	low	1—Sitting for 1 h and 30 min of treadmill walk and sitting uninterrupted for 6.5 h 2—Sitting for 1 h and 30 min of treadmill walk and every consecutive 30 min sitting and 3 min treadmill walk	8 h	Uninterrupted sitting for 8 h	Not administered	Morning exercise reduces BP during a period of 8 h in older overweight/obese adults compared with prolonged sitting. Combining exercise with regular breaks in sitting may be of more benefit for lowering BP in women than in men.
Wheeler, M.J.; Green, D.J.; Ellis, K.A.; Cerin, E.; Heinonen, I.; Naylor, L.H.; Larsen, R.; Wennberg, P.; Boraxbekk, C.J.; Lewis, J.; Eikelis, N.; Lautenschlager, N.T.; Kingwell, B.A.; Lambert, G.; Owen, N.; Dunstan, D.W., 2020, Australia	“To compare the effects of morning bout of moderate intensity exercise with and without light intensity walking” [26]	N = 65 Participants are adults with history of obesity	Exercise	Low and moderate	Iight intensty walking during the work/office time with moderate intensity exercise in the morning sessions	3 min of walking for every 30 min of SB	1 h Morning exercise and uninterrupted sitting for 8 working/office hours	Nill	Increase in working memory net AUC z score Increase in serum BDNF (brain derived neurogenic factor)
Stephens, S.K.; Eakin, E.G.; Clark, B.K.; Winkler, E.A.H.; Owen, N.; LaMontagne, A.D.; Moodie, M.; Lawler, S.P.; Dunstan, D.W.; Healy, G.N., 2018, Australia	To check the efficiency of Stand up Victoria intervention-arm techniques to break SB and promote PA [27,28]	N = 134 Participants are desk bound workers	Sit and stand workstation	Low	Standing and walking	Standing or walking for every 30 min of sitting	Baseline data collected using ActivPAL-3	Nill	Decrease in SB Increase in PA by 3 h a day during 8 h of work time
Carter, S.E.; Draijer, R.; Holder, S.M.; Brown, L.; Thijssen, D.H.J.; Hopkins, N.D., 2018, The Netherlands	“To study the acute CBF responses in SB and assess the cerebrovascular effects of breaking up prolonged sitting” [28]	N = 15 Participants were deskbound office workers	Walking	Low	Treadmill walking	2 min of walking for every 30 min in 4 h of prolonged sitting 4 min of walking for every 120 min in 8 h of sitting	4 h of prolonged sitting	Nill	Increase in middle cerebral artery velocity in 2 min walk as compared to 4 h sitting Increase in cerebral auto regulation
Antle, D.M.; Cormier, L.; Findlay, M.; Miller, L.L., 2018, Canada	“To compare the changes in lower extremity discomfort, blood pressure and blood flow accumulation during a light load repetitive upper limb work task in seated and standing posture” [29]	N = 16 Adult participants who are desk bound	Light load box folding	Low	Standing	34 min of standing and sitting while doing the activity	Baseline data collected pre intervention	Nill	Increase in mean blood flow by 77% in alternated sitting and standing Decreased lower limb discomfort
Cooper, A.R.; Moore, L.A.; McKenna, J.; Riddoch, C.J., 2000, UK	“To investigate the effects of 6 weeks program of moderated intensity exercise on day time ambulatory blood pressure among unmediated hypertension office workers” [30]	N = 90 Adult participants who are desk bound workers	Exercise	Moderate	Exercise set	6 weeks	Participants who are not induced to any exercise other than the usual activity	Nill	decrease in daytime ambulatory blood pressure with 2.8 mmHg in systolic and 1.9 mm

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
