# Peer review of "Influence of Sedentary Behaviour Interventions on Vascular Functions and Cognitive Functions in Hypertensive Adults—A Scoping Review on Potential Mechanisms and Recommendations"

_ijerph, 2022, doi:10.3390/ijerph192215120_

Round 1
Reviewer 1 Report
After I reviewed this paper, I believe that the paper will be useful for those who interested in the effectiveness of intervention/break of sedentary behavior on vascular & cognitive functions in HT. However, I have some suggestions as the follows.
1. Eventhough the PA, SB and NMD are common used as the abbreviation, I suggest to state the full name with (abbreviation) prior to mention the abbreviation in the text and table. (LINE96, LINE117, Table4)
2. LINE 80: I suggest to revise the definition of sedentary behavior based on the ref No.9; SB defined as any waking behavior that requires low energy expenditure (≤ 1.5 MET) such as prolonged sitting, reclining or lying down. Then this definition will distinguish from the physical inactivity.
3. Table4, check the author name and year of publication of Reference No. 21 (Aguirre-Betolaza AM et al., in 2020)
4. Please check and revise the reference format/style
Author Response
We are grateful for the valuable reviews provided by the peer review team. We have made the necessary changes and highlighted the same. We have provided a detail changes in the table below for your reference
|
Reviewer |
SL.NO |
Suggestions received |
Clarifications |
Line number |
|
1 |
1 |
Eventhough the PA, SB and NMD are common used as the abbreviation, I suggest to state the full name with (abbreviation) prior to mention the abbreviation in the text and table. (LINE96, LINE117, Table4) |
The suggested changes has been incorporated |
121 122
|
|
2 |
LINE 80: I suggest to revise the definition of sedentary behavior based on the ref No.9; SB defined as any waking behavior that requires low energy expenditure (≤ 1.5 MET) such as prolonged sitting, reclining or lying down. Then this definition will distinguish from the physical inactivity. |
The suggested changes has been incorporated The information on MET has been highlighted |
83 |
|
|
3 |
Table4, check the author name and year of publication of Reference No. 21 (Aguirre-Betolaza AM et al., in 2020) |
The suggested changes has been incorporated
|
Table 4 |
|
|
4 |
Please check and revise the reference format/style |
The suggested changes has been incorporated
|
312 |
Reviewer 2 Report
Congratulations for your manuscript. I have a suggestion about table 4 "Increased in PA from 188 min to 117 min". I just suggest to review this data.
Author Response
We are grateful for the valuable reviews provided by the peer review team. We have made the necessary changes and highlighted the same. We have provided a detail changes in the table below for your reference
|
2 |
1 |
I have a suggestion about table 4 "Increased in PA from 188 min to 117 min". I just suggest to review this data |
The suggested changes has been incorporated
|
Table 4 |
Reviewer 3 Report
While applauding the authors on their efforts in producing this study, the fact is that by reading the abstract it is possible to understand significant scientific and methodological flaws.
Scientific flaws:
It begins by stating “Since the workplace has become desk-based and leisure time has become dominated by digital relaxation modes, the world is moving at a pace where physical activity has become a time-bound routine” and then searches back to 1964. However, there were no problems with sedentary lifestyles and, as a result, the diseases associated with them in the 1960s for a variety of reasons, the most immediate being the common citizen's lack of access to cars, computers, and mobile phones. Only its availability in most wealthy nations, resulting in people not having to move or not having to do so on their own, is what led to the development of such diseases, which are still not manifest in the third world. As a result, aside from the theoretical framework, it adds nothing to current understanding, does not frame the problem, and does not indicate the study's purpose.
Methodological flaws:
-It is unacceptable that 20,625 articles (respecting 58 years old) are discovered and only 12 fit the standards, 11 of which are from the past 10 years and 9 from the previous 5 years. This suggests that rigorous inclusion/exclusion criteria were not set, and the search equation(s) did not contain the relevant terms for the proposed study.
-The authors claim to have followed the grid of Arksey H, O'Malley L. (2005), yet this work is about 20 years old, and the authors simply advise measures to take. There is a set of methodical and scientific processes that are absolutely important inside these stages, which did not occur here.
-The search equation(s) are not properly stated, and we are given a table (no2) with an infinite number of terms, which is why more than 20 thousand articles were obtained and virtually all of them were rejected at the outset.
- The inclusion/exclusion criteria merely mention the time span; nothing about the language or languages in which the articles are produced is included; whether they are merely articles or all of the papers discovered; if they must be open access; whether they must contain an abstract and full text; the kind of journal (indexed with IF or not); the type of review (peer or not), and so on.
-The fact that none of this was employed in the model's identification phase, along with the lack of well-structured research equations, resulted in the large number of articles. That is to say, the filters were not used correctly.
- On the other side, during the screening phase, 20189 documents were rejected based on three "criteria," although the issue of how that selection was done remains unanswered: Have all of the entire texts been read? in less than two months? This would result in you receiving over 250 papers every day. Did just the abstracts get read? Were they chosen based on titles? Were they chosen based on the keywords?
- etc.
In short, the paper has severe flaws and is not of sufficient scientific quality.
Author Response
We are grateful for the valuable reviews provided by the peer review team. We have made the necessary changes and highlighted the same. We have provided a detail changes in the table below for your reference
|
3 |
1 |
It begins by stating “Since the workplace has become desk-based and leisure time has become dominated by digital relaxation modes, the world is moving at a pace where physical activity has become a time-bound routine” and then searches back to 1964. However, there were no problems with sedentary lifestyles and, as a result, the diseases associated with them in the 1960s for a variety of reasons, the most immediate being the common citizen's lack of access to cars, computers, and mobile phones. Only its availability in most wealthy nations, resulting in people not having to move or not having to do so on their own, is what led to the development of such diseases, which are still not manifest in the third world. As a result, aside from the theoretical framework, it adds nothing to current understanding, does not frame the problem, and does not indicate the study's purpose. |
The timeline was set to 1964 since there were articles in our database search that dated back to 1960 and had title matches based on hypertension and healthy ageing. But those articles had to be excluded due to lack of full text and did not focus on the current topic. |
|
|
|
2 |
It is unacceptable that 20,625 articles (respecting 58 years old) are discovered and only 12 fit the standards, 11 of which are from the past 10 years and 9 from the previous 5 years. This suggests that rigorous inclusion/exclusion criteria were not set, and the search equation(s) did not contain the relevant terms for the proposed study |
Since our topic focuses on hypertension and its adverse effects as a baseline search, we had 20,625 articles in the initial search. Later on when our focus was narrowed to SB and PA based articles, the number of articles go narrowed down.
The exclusion and inclusion criteria has been mentioned and highlighted |
126-130 |
|
|
3 |
The authors claim to have followed the grid of Arksey H, O'Malley L. (2005), yet this work is about 20 years old, and the authors simply advise measures to take. There is a set of methodical and scientific processes that are absolutely important inside these stages, which did not occur here |
According to of Arksey H, O'Malley L. (2005). There are 5 stages Stage 1. Identifying the research question Stage 2 Identifying relevant studies Stage 3 Study selection Stage 4 Charting the data Stage 5 Collating, summarising and reporting the results. https://www.tandfonline.com/doi/abs/10.1080/1364557032000119616
Stage 1 Stage 2 Stage 3 Stage 4 Stage 5 |
112 117 119 131 Table 4 |
|
|
4 |
The search equation(s) are not properly stated, and we are given a table (no2) with an infinite number of terms, which is why more than 20 thousand articles were obtained and virtually all of them were rejected at the outset |
The MeSH terms were confined to the table 2 but the criteria for rejection was either the article did not specify the need for a physical activity or did not highlight the hypertension population |
|
|
|
5 |
The inclusion/exclusion criteria merely mention the time span; nothing about the language or languages in which the articles are produced is included; whether they are merely articles or all of the papers discovered; if they must be open access; whether they must contain an abstract and full text; the kind of journal (indexed with IF or not); the type of review (peer or not), and so on |
The suggested changes has been incorporated
|
126-130 |
|
|
6 |
The fact that none of this was employed in the model's identification phase, along with the lack of well-structured research equations, resulted in the large number of articles. That is to say, the filters were not used correctly |
The clarifications has been highlighted |
126-131 |
|
|
7 |
On the other side, during the screening phase, 20189 documents were rejected based on three "criteria," although the issue of how that selection was done remains unanswered: Have all of the entire texts been read? in less than two months? This would result in you receiving over 250 papers every day. Did just the abstracts get read? Were they chosen based on titles? Were they chosen based on the keywords? |
Out of 20189, it was 436 articles were screened for full test, rest of them were exclude based on the title , abstract and keywords. |
Fig 1 |